# Reliability of the Polar Vantage M Sports Watch when Measuring Heart Rate at Different Treadmill Exercise Intensities

**DOI:** 10.3390/sports8090117

**Published:** 2020-08-23

**Authors:** Mike Climstein, Jessica L. Alder, Alyce M. Brooker, Elissa J. Cartwright, Kevin Kemp-Smith, Vini Simas, James Furness

**Affiliations:** 1Clinical Exercise Physiology, School of Health and Human Sciences, Southern Cross University, Bilinga, QLD 4225, Australia; 2Physical Activity, Lifestyle, Ageing and Wellbeing, Faculty Research Group, Faculty of Health Sciences, The University of Sydney, Lidcombe, NSW 2006, Australia; 3Water Based Research Unit, Bond University, Robina, QLD 4226, Australia; vsimas@bond.edu.au (V.S.); jfurness@bond.edu.au (J.F.); 4Bond Institute of Health and Sport, Faculty of Health Sciences and Medicine, Robina, QLD 4226, Australia; jessica.alder@student.bond.edu.au (J.L.A.); alyce.brooker@student.bond.edu.au (A.M.B.); elissa.cartwright@student.bond.edu.au (E.J.C.); kkempsmi@bond.edu.au (K.K.-S.)

**Keywords:** wearable technology, wearable device, smart wristband, photoplethysmography, heart rate response, aerobic exercise, monitoring

## Abstract

Background: Usage of wrist-worn activity monitors has rapidly increased in recent years, and these devices are being used by both fitness enthusiasts and in clinical populations. We, therefore, assessed the test–retest reliability of the Polar Vantage M (PVM) watch when measuring heart rate (HR) during various treadmill exercise intensities. Methods: HR was measured every 30 s (simultaneous electrocardiography (ECG) and PVM). Test–retest reliability was determined using an intraclass correlation coefficient (ICC) with 95% confidence intervals (CIs). Standard error of measurement (SEM) and smallest real difference (SRD) were used to determine measurement variability. Results: A total of 29 participants completed the trials. ICC values for PVM during stages 1, 2 and 5 demonstrated good to excellent test–retest reliability (0.78, 0.78 and 0.92; 95% CI (0.54–0.90, 0.54–0.9, 0.79–0.97)). For PVM during stages 0 (rest), 3 and 4, the ICC values indicated poor to good reliability (0.42, 0.68 and 0.58; 95% CI (−0.27–0.73, 0.32–0.85, 0.14–0.80)). Conclusion: This study identified that the test–retest reliability of the PVM was comparable at low and high exercise intensities; however, it revealed a poor to good test–retest reliability at moderate intensities. The PVM should not be used in a clinical setting where monitoring of an accurate HR is crucial to the patients’ safety.

## 1. Introduction

Recent advances in wrist-worn technology have led to the development and distribution of many wrist-worn activity monitors (AMs), with 140 million devices projected to be sold by 2022 [1]. The non-invasive technology used in wrist-worn AMs utilises photoplethysmography (PPG), which uses optical measurement to detect volumetric changes in blood circulation under the skin [2]. With the increased demand and usage of wrist-worn AMs, it is expected that a large proportion of the population will turn to these devices to monitor various health related measures, including heart rate (HR). As these devices become more prevalent in the general population, so does their use in a clinical setting, making it imperative to investigate the reliability of these devices against comparable medical grade technology.

Previously, wearable chest strap devices with an accompanying wristwatch which uses telemetry, such as the Polar T31 chest strap (Polar, Kempele, Finland), were shown to be a reliable measure of HR compared to gold-standard electrocardiography (ECG) [3]. There are a multitude of studies which have investigated the validity of a wide range of wrist-worn AMs including Apple Watch 1, Fitbit Charge HR, Garmin Forerunner 225, Garmin Vivosmart HR+ and TomTom Cardio [4,5,6,7,8]; however, this is not reflected in the literature with regard to the devices’ reliability.

While the number of studies performed to date has been limited, two recent studies demonstrate that AMs display good test–retest reliability for measuring HR at rest and during exercise. This includes the Apple Watch [9] and Fitbit Charge HR [10], for monitoring HR. Khushhal et al. [9] assessed the inter and intradevice reliability of the Apple Watch during exercise and recovery. They reported the intradevice reliability to be good during walking and in recovery, and reliability improved with increased exercise intensity. The interdevice reliability was found to be very good with low standardised typical errors and a good to very good intraclass correlation coefficient (ICCs, 0.82–1.00). Similarly, Nazari et al. [10] reported the Fitbit Charge HR to have excellent reliability over three conditions—rest, exercise and recovery.

Currently, only one study has used the Polar Vantage M (PVM) to investigate its accuracy when estimating energy expenditure (EE) when compared to indirect calorimetry during various activities ranging from sitting to playing football [11]. Researchers found the PVM to have good to moderate accuracy in estimating EE; however, the level of accuracy was activity-dependent (e.g., sitting, mopping, walking, jogging, weight training, stationary cycling and floorball). This study, though, did not assess the reliability of the PVM in measuring HR during exercise. As such, the reliability of the PVM in accurately measuring HR during exercise is unknown. The purpose of this study was to assess the test–retest reliability of the PVM to measure HR during various treadmill intensities.

## 2. Materials and Methods

### 2.1. Design and Participants

The study was a cross-sectional observational design and was approved by the Bond University Human Research Ethics Committee (ML01928). The test–retest reliability of the PVM was established through two separate testing sessions, held at least 24 h apart. The same researcher was used to record the HR measured on the PVM throughout the study to ensure standardisation, and a second researcher recorded the HR measured on the ECG monitor throughout the study.

Healthy participants aged 18 to (enter age of your oldest participant) were recruited by means of electronic and printed posters, which were distributed via social media platforms and within the university premises. Previous research has shown that a sample size of at least 15–20 was considered an adequate sample size to conduct reliability studies which collected continuous data [12].

Inclusion criteria: Healthy male and female participants, 18 years or older who provided consent to undertake the research protocol, were English speaking, and completed the Exercise and Sports Science Adult Pre-Exercise Screening System (APSS) with no contraindications (relative or absolute) to exercise were eligible to participate.

Exclusion Criteria: Subjects who identified any pre-existing respiratory, cardiovascular or metabolic conditions, were under 18 years of age, taking medications that affect HR, were pregnant, had musculoskeletal injuries currently or within the last six weeks (e.g., back pain, ankle sprain or osteoarthritis), or refused to provide informed consent to perform maximal treadmill exercise were excluded from the study.

### 2.2. Measures

#### 2.2.1. Wrist-Worn Activity Monitor

The Polar Vantage M (Polar Electro, Kempele, Finland) is a combined activity and multi-sport watch that measures HR peripherally via PPG [13]. The technology allows HR to be assessed with minimal interference from motion artifacts. The laboratory temperature and humidity were tightly controlled to lessen environmental artifact. Additional functions of the PVM include step counter, training load, training programs, sports profiles, GPS and altitude. For the purpose of this study, the “running” profile was selected to track HR.

#### 2.2.2. Cardiac Stress Testing System

Electrocardiography is considered the gold standard for measuring HR; therefore, the real-time ECG (standard 3 lead plus V5) was used in this study to measure HR [6]. The stress testing system also included the function to store, print and export wireless ECG data and filter noise to maintain high ECG integrity [14]. The wireless system was used due to its small size and ability to be worn independently on the participant’s waist. The five electrodes were placed on the RA, LA, LL, RL and V5 positions (Figure 1). The stress testing system’s exercise protocol function was set to graded exercise test (Bruce treadmill protocol selected) to provide real-time monitoring of each stage in the protocol. Additionally, the treadmill is controlled by the stress testing system to automatically increase the treadmill speed and grade.

#### 2.2.3. Bruce Protocol Treadmill Stress Test

The protocol involved three-minute continuous stages increasing speed and elevation until the participant reached volitional exhaustion [15], therefore allowing the participant’s HR to work progressively from low to high intensities; however, as each stage is three minutes in duration, this allowed each participant the opportunity to reach steady state with regard to HR. In Table 1, we have estimated the relative (mL/kg/min) oxygen consumption for each of the stages (PVO_2_). Stages 1 and 2 represent the lowest PVO2, while stages 5 and 6 represent the highest PVO_2_. The test was terminated at volitional exhaustion or when participants reached their age-predicted HR maximum (APHRM, i.e., HRmax = 220-age) [16]. The stress test system continually provided the percentage to APHRM throughout the test on the ECG monitor. The Bruce Protocol was chosen over other protocols as it allows participants to reach a steady state and maximal exertion in 20 min or less [17].

#### 2.2.4. Laboratory Conditions

The environment in the laboratory was standardised throughout all sessions by maintaining an ambient temperature of 21 degrees Celsius (69.8 °F) with use of a fan in front of the participant to minimise thermoregulatory strain and facilitate convective cooling during the maximal exercise testing.

Anthropometric and sex data were extracted to characterise participants and determine body mass index (BMI, kg/m^2^). Age-predicted HR max was determined prior to the start of each test to ensure participants’ safety during the maximal exercise, thereby allowing the researchers to terminate the test when age-predicted maximal HR was reached.

### 2.3. Procedure/Protocol

Participants attended the laboratory on two separate occasions, 48 h apart. Participants were advised to refrain from exercise two days prior to both testing occasions. The first day of testing required participants to complete the pre-activity screening form and sign an informed consent. Initially, height and mass were measured and recorded. ECG electrodes were then placed on participants in a standardised manner [19] including the limb leads (I, II, III, AvR, AvL, AvF) and V5 positions (Figure 1). Participants were then instructed to sit on a chair while the ECG leads were attached and the PVM placed on the right wrist, with the number of notches remaining on the wrist band recorded to ensure standardisation in the following testing session. Each participant was informed that the test would be terminated if they experienced any dizziness, chest pain, extreme shortness of breath, light-headedness and/or nausea or if an abnormal HR, blood pressure or arrhythmia was observed (i.e., symptom limited).

To determine the test–retest reliability of the PVM, HR measurements were taken every 30 s throughout the Bruce Protocol, including a three-minute resting period while sitting in a chair, a two-minute resting period while standing upright on the treadmill and a cool-down period.

### 2.4. Statistical Analysis

Statistical analyses were performed using SPSS (Version 25.0; IBM Corp, Armonk, NY, USA). Normality of the data was assessed by investigating kurtosis (visually and calculated), skewness (visually and calculated), Q–Q plots, as well as the Kolmogorov–Smirnov test with the Lilliefors significance correlation. Significant differences between genders were determined using an independent-samples *t*-test. Test–retest reliability was determined using an intraclass correlation coefficient (ICC) and associated 95% confidence intervals (CIs) [20]. ICC estimates were calculated using a two-way mixed-effects model with absolute agreement and were used to assess test–retest reliability, as outlined by Koo and Li [21]. Fleiss and colleagues [22] have suggested that ICC values < 0.40 indicate fair poor reliability, 0.40 to 0.59 indicate fair reliability, 0.60 to 0.74 indicate good reliability and an ICC value greater than 0.75 indicates excellent reliability.

To assess measurement variability, the standard error of measurement (SEM) was calculated. This measurement is defined by the equation SEM = SD √(1 − r), with r representing the ICC [23]. Standard error of measurement was calculated to provide an “absolute index of reliability” associated with a measurement [20]. The following criteria were used for absolute measures: good SEM = <10 bps and poor SEM = ≥10 bps [24,25]. To determine any clinically important changes, the smallest real difference (SRD) was used and calculated using the equation SRD = 1.96 × SEM × √2 [26].

The level of agreement (LoA) between the measurements recorded on the PVM on days 1 and 2 during different stages of the Bruce protocol is illustrated through Bland–Altman plots, with the corresponding 90% LoA, using the formula: mean difference between measures ±1.645 × SD [27].

## 3. Results

### 3.1. Sample

One participant was excluded from the final data analyses due to an ECG arrhythmia during testing. Therefore, a total of 29 participants successfully completed the study and were included in the data analyses. Males and females were of similar age; however, males were significantly taller (*p* < 0.05, +5.9%), higher mass (*p* < 0.05, +25.6%) and a higher BMI (*p* < 0.05, +13.6%). With regard to BMI, the majority of females had a normal BMI (≥18.5 to ≤24.9kg/m^2^, 69%), while the majority of males were overweight (25.0 to ≤29.9 kg/m^2^, 84.6%), no participants were underweight (<18.5 kg/m^2^) or obese (≥30.0+ kg/m^2^). There was no difference between sex with regard to resting HR (*p* = 0.60) or maximal exercise HR (*p* = 0.84). As there was no difference with regard to age between genders, resting HR or HR max, all participant data were combined (Table 2).

There were no significant differences seen with regard to HR determined by ECG days 1 to 2 across all stages (*p* = 0.151 to 0.885) or in the PVM days 1 to 2 (*p* = 0.124 to 0.885) (Table 3). The greatest day 1 to day 2 difference with the ECG was seen at rest (mean 1.8 bpm), whereas the greatest difference in the PVM was seen during stage 4 (mean 5.7 bpm). Table 3 illustrates the HRs seen over both testing sessions with the ECG and PVM.

### 3.2. Reliability Analysis

#### Test–Retest Reliability of ECG and Polar Vantage M Heart Rate Variables

Interclass correlation coefficient values for ECG data were excellent for all stages of the Bruce protocol ranging from 0.89 to 0.99 (Table 3).

ICC values for PVM during stages 1, 2 and 5 showed excellent reliability to good reliability [21] (0.78, 0.78 and 0.92; 95% CI (0.54–0.90, 0.54–0.9, 0.79–0.97)). For PVM during stages 0, 3 and 4, the ICC values indicate poor–excellent reliability (0.42, 0.68 and 0.58; 95% CI [−0.27–0.73, 0.32–0.85, 0.14–0.80]). See Table 3 for further details.

In terms of absolute measures, SEM and SRD values for both the ECG and PVM are presented in Table 4 and Table 5.

### 3.3. Limits of Agreement

A Bland–Altman plot was produced to represent the 90% LoA between the HR measurements recorded during stages 1, 4 and 5 on days 1 and 2 (Figure 2, Figure 3 and Figure 4). During stage 1, the mean difference was 1.31 bpm, with the upper and lower LoA being 17.76 bpm and −15.14 bpm. During stage 4, the mean difference was 5.6 bpm, with the upper and lower LoA being 31.71 bpm and −20.51 bpm. During stage 5, the mean difference was −0.87 bpm, with the upper and lower LoA being 11.4 and −13.14 bpm. Figure 3 shows the majority of data points being within the LoA, ranging from −12.5 to 13.5 bpm (26). Figure 4 shows most data points being within the LoA, ranging from −20.50 to 31 bpm (51.5). Figure 5 shows only one data point being outside of the LoA. Data points within the LoA range from −13.00 to 11.00 bpm (24).

## 4. Discussion

The purpose of this study was to assess the test–retest reliability and agreement of the PVM when measuring HR during incremental exercise on a treadmill as compared to the gold standard ECG. The main findings of this study indicate the PVM has excellent to fair reliability at low and high intensities, with poor–good reliability at moderate intensities. The LoA, however, show a wide range for the PVM HR measurements over two exercise sessions and varied according to the exercise intensity.

In accordance with the American National Standard of “cardiac monitors, heart rate meters, and alarms”, an acceptable error can range from two percent at maximal levels to 10% in resting conditions [28]. Given the aforementioned recommendations and large limits of agreement of the PVM, displaying a range of up to 51 bpm difference to actual HR, the authors recommend that PVM it is not acceptable for clinical use. For example, individuals with exertional angina at moderate to high exercise intensities would be at significant risk for imminent myocardial infarction if the exercise HR were not accurate and exceeded the anginal threshold [29,30].

### 4.1. Test–Retest Reliability

It was found that the PVM test–retest reliability had different results based upon the exercise intensities. We found the PVM has excellent to good reliability during stages 1, 2 and 5 of the Bruce Protocol, which correspond to low-intensity exercise and high-intensity exercise. During stages 0, 3 and 4, the study showed that the PVM had poor–good reliability.

In a 2017 randomised control trial by Maas and colleagues [31], they reported that novice runners demonstrated larger kinematic changes during a maximal exercise (11.88 km/h) running protocol compared to competitive runners. Novice runners displayed a significant increase in both peak trunk flexion (*p* < 0.05, 3.0°) and trunk rotation (*p* < 0.05, 3.5°). It is possible to extrapolate these data to assume novice runners would also demonstrate increased arm swing during high-intensity protocols due, in part, to the increase in trunk rotation. The current study showed a reduction in reliability during stages 3 and 4 of the protocol with an improvement in stage 5. It can be postulated that because stages 3 and 4 of the Bruce protocol resulted in higher heart rates, there was more arm movement by the participants who were novice runners which may account for increased motion artefact in the PVM during these stages which potentially could skew results. However, the more experienced, fitter participants were able to complete stage 5 of the treadmill protocol and thus less likely to have significant trunk movement. This is reflected in the higher reliability seen in this high-intensity exercise in these participants. Although the ICC values were high, the 95% CIs were found to be wide, which indicates greater variability; this was seen in stages 3 and 4 of the treadmill protocol.

It could be argued that the poor reliability in results is due to confounding factors within the Bruce protocol between testing sessions. However, if this was the case, then the ECG results would also have illustrated poor reliability. The test–retest reliability of the ECG was assessed to eliminate the possibility of the Bruce treadmill protocol being the cause of difference. However, the Bruce protocol has been previously shown to have high reliability with regard to HR and maximal oxygen consumption [32]. Additionally, the ECG was shown to have excellent test–retest reliability throughout all stages of the Bruce treadmill protocol, hence all exercise intensities.

### 4.2. Limits of Agreement

Bland–Altman plots were used to interpret the 90% LoA between the difference and the mean of the PVM HR readings for stages 1, 4 and 5. These stages represent low, moderate and high intensities of exercise, reflected by HR. Although the majority of the data points in were within the LoA, the variation of HR measured could be up to 26 bpm. Similarities were also seen in stage 5 (24 bpm); however, stage 4 revealed differences HR up to 51.5 bpm.

Due to the wide LoA obtained by the PVM, it was determined that the PVM is not an appropriate measure of HR within a clinical setting. During the Bruce protocol, the majority of the data points for each stage were within the LoA; however, these were found to be as wide as 51.5 bpm. As discussed by Kottner et al. [33], the acceptable difference between data points, or HR in this case, is not a statistical decision but rather a clinical decision. With this in mind, it may be appropriate for the PVM to be used to track HR and, therefore, exercise intensity in recreational or competitive athletes. However, as the difference in actual HR compared to what the PVM is displaying may be up to 51 bpm different, the authors suggest the PVM would not be advisable to use in monitoring HR in cardiac patients or any individuals with chronic diseases and/or conditions where overexertion may pose a risk to the individual’s wellbeing. However, the PVM could be utilised as a measure of HR for recreational or competitive athletes where the purpose is to solely track HR or exercise intensity, without incorporating the measurement of HR for any health-related purposes.

### 4.3. Strengths and Limitations

Strengths of this study include the use of ECG as the gold standard measure for comparing HR in wearable sensors; previous studies have also utilised ECG as the criterion measure [34,35,36]. Electrocardiography demonstrated a high level of reliability in determining HR over two days. Additionally, the use of the Bruce treadmill protocol, automated by the X-Scribe system, ensured precision (automated) application of changes to the treadmill speed and grade throughout both testing sessions. Lastly, we have a number of investigators responsible for specific aspects of data collection; their roles did not change between participants nor between testing occasions.

The researchers were not blinded throughout the study due to the need for each researcher to participate in the collection of data, and therefore, the study could be prone to bias. For example, our participants were a convenience sample of healthy, active university students; therefore, selection/sampling bias may have affected our findings. Further, these participants were over the age of 18 years (mean age of 26 years), which limits the application of our findings to other populations, such as middle-aged and those individuals with chronic conditions. Future studies should incorporate participants with different health concerns, a wider variety of ages, as well as varying levels of fitness to be able to more widely extrapolate the results.

Participants completed the Bruce protocol within 48 h of their previous session, and therefore, the results of their second session may have been influenced by fatigue or muscle soreness, which could in turn affect their performance. However, all participants completed the same level of stages in day 2 as in day 1, and their maximal exercise HR between the two sessions was non-significant and less than one beat per minute (0.526 bpm).

Additionally, this study only investigated treadmill walking, jogging and running, and therefore, our findings are not generalisable to other activities, such as cycling, various sports or occupational activities.

## 5. Conclusions

When measuring HR during exercise, the PVM was shown to have excellent to good test–retest reliability at low and high intensities, with poor–good test–retest reliability at moderate intensities. Due to the wide LoA between the two sessions, the current research findings suggest that the PVM does not measure HR to the same standard as ECG and therefore is not a suitable choice for measuring HR within a clinical population. Future research should investigate the use of PVM in participants with chronic conditions where these individuals will most likely have lower fitness levels, and exercise intensity may be limited to low to moderate.

## Figures and Tables

**Figure 1 sports-08-00117-f001:**
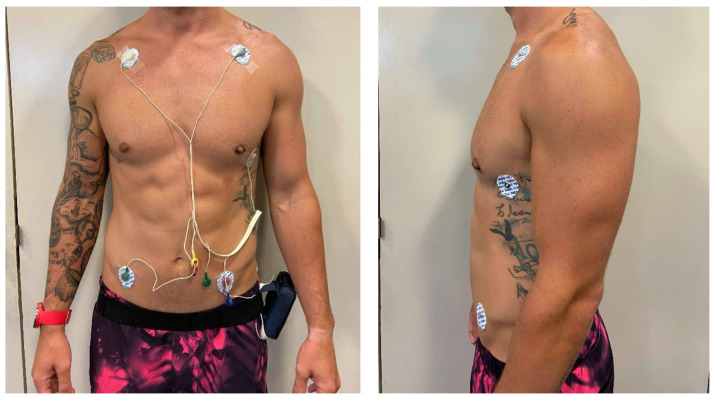
Participant shown with electrocardiography (ECG) configuration.

**Figure 2 sports-08-00117-f002:**
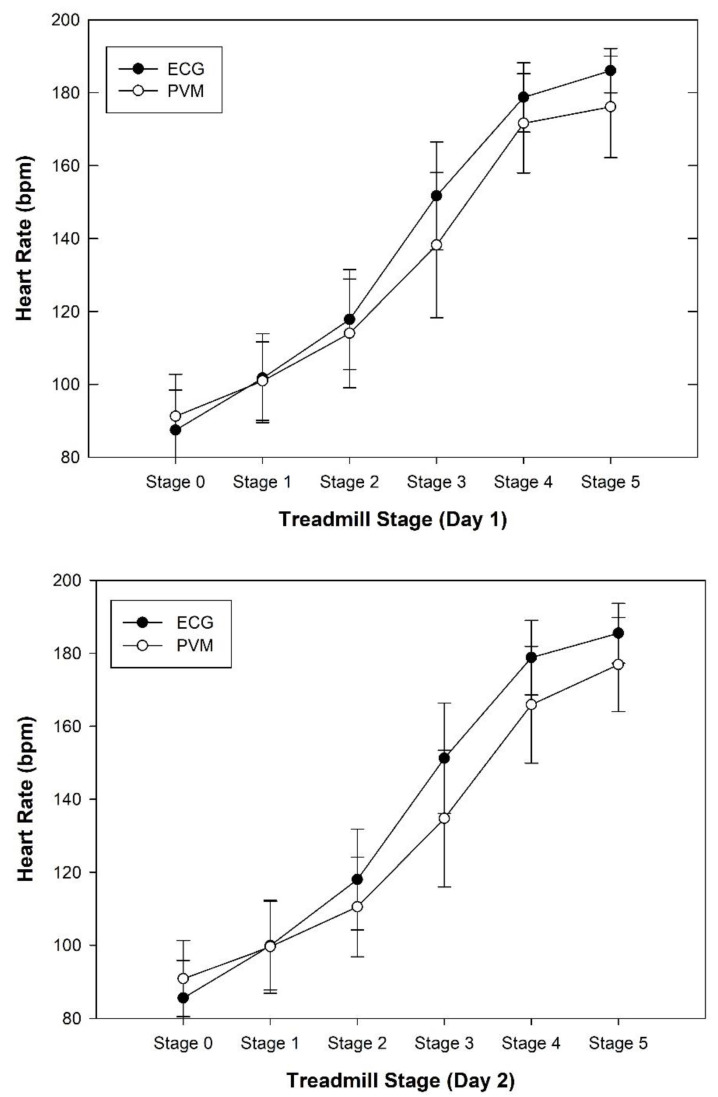
ECG and polar heart rate results on days 1 and 2.

**Figure 3 sports-08-00117-f003:**
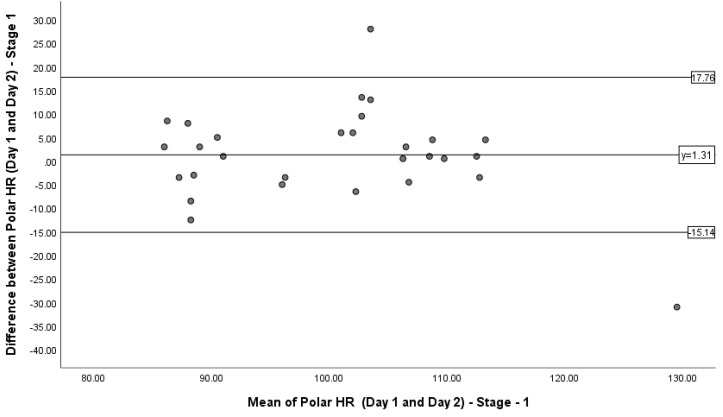
Bland–Altman plot representing the 90% limits of agreement between the difference in heart rate measurements on days 1 and 2 during stage 1 of the Bruce Protocol and the mean of the Polar Vantage M (PVM) heart rate readings on days 1 and 2 during stage 1 of the Bruce Protocol.

**Figure 4 sports-08-00117-f004:**
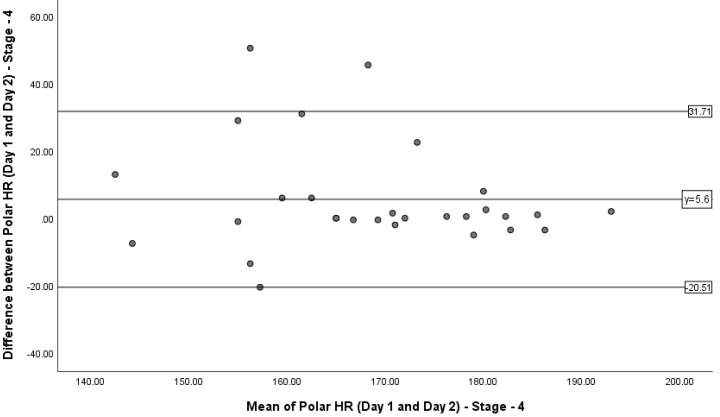
Bland–Altman plot representing the 90% limits of agreement between the difference in heart rate measurements on days 1 and 2 during stage 4 of the Bruce Protocol and the mean of the PVM heart rate readings on days 1 and 2 during stage 4 of the Bruce Protocol.

**Figure 5 sports-08-00117-f005:**
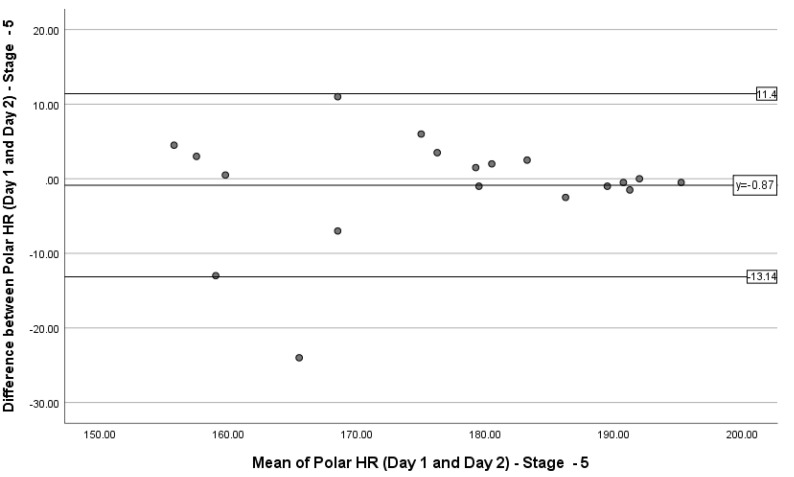
Bland–Altman plot representing the 90% limits of agreement between the difference in heart rate measurements on days 1 and 2 during stage 5 of the Bruce Protocol and the mean of the PVM heart rate readings on days 1 and 2 during stage 5 of the Bruce Protocol.

**Table 1 sports-08-00117-t001:** Bruce protocol treadmill [18], where PVO_2_ is predicted relative oxygen consumption.

Stage	Minutes	Speed (km/h)	Grade (%)	PVO_2_ (mL/kg/min)
1	3	2.7	10	17.5
2	6	4.0	12	24.5
3	9	5.4	14	35.0
4	12	6.7	16	45.5
5	15	8.0	18	52.5
6	18	8.8	20	63.0

**Table 2 sports-08-00117-t002:** Participant characteristics.

	Females (*n* = 13)				Males (*n* = 16)			
	Mean	± SD	Min	Max	Mean	±SD	Min	Max
Age (years)	26	3.85	22.00	33.00	26.25	3.17	23.00	35.00
Height (m)	1.70	0.05	1.65	1.79	1.79 *	0.06	1.70	1.88
Mass (kg)	64.95	6.50	52.40	77.5	81.85 *	8.73	67.5	94.7
BMI (kg/m^2^)	22.50	2.07	19.25	26.26	25.54 *	2.54	20.83	28.06
Rest HR (bpm)	72.4	7.6	61	87	70.4	11.9	45	90
Max HR (bpm)	194.0	3.2	187	198	193.8	3.2	185	197

SD = standard deviation, Min = minimum, Max = maximum, m = meters, kg = kilograms, BMI = body mass index, Max HR = maximum heart rate. * = *p* < 0.05.

**Table 3 sports-08-00117-t003:** ECG and Polar Vantage M heart rate treadmill protocol results. Values are bpm, mean (± SD).

Stage	ECG: Day 1	ECG Day 2	ECG: Difference	PVM: Day 1	PVM: Day 2	PVM: Difference
Rest	87.4 (11.0)	85.6 (10.3)	1.8	91.2 (11.4)	90.9 (10.4)	0.3
1	101.7 (12.7)	99.9 (12.1)	1.6	100.9 (10.8)	99.6 (12.7)	1.3
2	117.9 (13.7)	118.1 (13.7)	0.2	114.0 (14.9)	110.6 (13.6)	3.4
3	151.6 (14.8)	151.3 (15.1)	0.3	138.2 (19.9)	134.7 (18.7)	3.5
4	178.7 (9.5)	178.8 (10.2)	0.1	171.6 (13.6)	165.9 (15.9)	5.7
5	186.0 (6.1)	185.5 (8.2)	0.5	176.1 (13.9)	176.9 (12.9)	0.8

**Table 4 sports-08-00117-t004:** Test–retest reliability of ECG and Polar Vantage M throughout Bruce Protocol.

Stage	Mean Difference	SEM	ICC	95% CI	SRD	% Difference	SEM	ICC	95% CI	SRD	% Difference
0	1.85	3.65	0.89	0.76–0.95	10.08	0.36	8.71	0.42	0.27–0.73	24.07	26.43
1	1.79	3.65	0.91	0.81–0.96	10.08	1.3	5.07	0.78	0.54–0.9	14.01	13.97
2	0.24	4.54	0.89	0.76–0.95	12.55	3.5	6.98	0.78	0.53–0.9	19.29	17.18
3	0.4	2.57	0.97	0.94–0.99	7.10	3.5	11.24	0.68	0.32–0.85	31.06	22.76
4	0.14	0.95	0.99	0.97–0.99	2.63	5.6	8.84	0.58	0.14–0.8	24.43	14.47
5	0.53	1.62	0.93	0.82–0.97	4.48	0.87	3.93	0.92	0.79–0.97	10.86	6.15

CI = confidence interval; ICC = intraclass correlation coefficient; SEM = standard error of measurement, SRD = smallest real difference.

**Table 5 sports-08-00117-t005:** Overall test–retest reliability of ECG and Polar Vantage M.

	ECG					Polar Vantage M				
Mean Difference	SEM	ICC	95% CI	SRD	% Difference	SEM	ICC	95% CI	SRD	% Difference
0.71	3.84	0.994	0.991–0.995	10.61	7.92	2.43	7.02	0.96	0.94, 0.97	19.40

CI = confidence interval; ICC = intraclass correlation coefficient; SEM = standard error of measurement, SRD = smallest real difference.

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
