# Peer review of "Reliability of the Polar Vantage M Sports Watch when Measuring Heart Rate at Different Treadmill Exercise Intensities"

_sports, 2020, doi:10.3390/sports8090117_

Round 1

Reviewer 1 Report

The paper deals with evaluation of reliability of a Polar wearable, an activity monitor measuring heart rate. The Polar device was evaluated against the, as the authors call it, “the gold standard” for HR measurements – ECG. But the ECG device’s metrological quality itself may vary, so the reader would expect the brand and the model of the ECG device and not only a description (e.g. of number of leads).

The authors should be aware that numbers and units, or numbers and % sign should be written separated by a single space (not 12%, but 12 % - see ISO 80000 standard) – correct throughout the text (e.g. line 22, line 167, line 259 etc).

Line 26, line 168, line 179, line 195- Bruce protocol – in the introduction the Bruce protocol should be explained a bit better – the stages of the protocol (in Table 1) are first described (low, moderate, high etc) very late in the text (line 195). Actually, the first time stages 1, 4 and 5 are described as low, moderate and high intensity is in line 257!! The authors should reorder the description (Stages should be described within the Method section.)

Line 89 – PPG is a method prone to extraneous interferences and moving artefacts. Novel PPG methodology deals with this problem, but “HR to be assessed“ completely “without interference” is technically quite impossible, so I would suggest the authors to change the word “without” with something less strong. Another quite strong error source is the skin temperature (and environment temperature).

Table 1 - what does Grade mean – is this the slope/elevation of the treadmill?

Line 106, line 177, line 118 – the readers would benefit from a description or at least a reference for APHRM – additionally, the APHRM seems a safety factor/criteria and I do not see huge relevance with the paper’s topics

Paragraph 112 – Apart air temperature the air relative humidity and even lighting conditions (PPG is a photosensitive method and the lux value is sometimes necessary to be controlled) should sometimes be monitored or controlled. Did the authors consider that?

Line 131 – why did the authors decide on 30 sec discrete intermittent measurements and not continuous ones?

Line 156, line 157, line 160 etc – (p<0.05) – is this piece of information result of a T test? If so, it should be added in section 2.4 and explained how was the data normality checked.

Line 163 – according to the SI units system meters should be written as “m”

Table 3 – ECG and PVM sections are missing.

Table 3 and table 4 – if the authors decided on including all these parameters, they should also be interpreted in the results section (e.g. what does SEM 3.84 versus SEM 2.43 actually mean? what does Mean difference in Table 4 mean? etc).

Line 222 – include the ANSI standard number in references.

Table 2 – in the Results section - How do the authors interpret that SD of both ECG and PVM are similar? PPG method should inherently be less constant that ECG method. If the SD is similar, the PVM is quite comparable to ECG measurements.

Line 280 – the fact that the Day 1 and Day 2 were 48 h apart is written at the end of the paper is very confusing for reader. This should be in the protocol section in the Methods section.

Line 153 Sample – were the participants more or less physically fit? The SD in Table 2 could be highly dependent on the level of fitness of the participants (more physically heterogeneous group would have larger SD values)

Line 29 – in my opinion the statement “PVM was reliable at low…” is quite strong, more adequate would be that it is comparable to ECG measurements

My main concern is that the authors (and some papers in the references) consider healthy participants to be physiologically constant – if the reliability is calculated from differences between two sessions (days), it is very important that the participants perform constantly in both days. Which is quite difficult considering the natural physiological changes, individual conditions, their level of fitness and activities during and between these days. To evaluate a novel device I think a more valuable information would be about the difference of the device to a reference device. If we are evaluating a device (Polar) using a reference device (ECG monitor), one of the interesting parameters could be the measuring error (device’s reading minus reference’s reading), which indicates the deviation of the device when compared to a reference device. This might be an interesting information on measuring quality of Polar versus ECG. In table 2, measuring errors could easily be included (i.e. day 1 - difference between PVM and ECG).

In conclusions the authors decide that PVM is suitable for HR measurements at certain levels of activity and not suitable in others. What were the decision criteria for this distinction? E.g. if the differences were larger than 50 bpm, the device is not suitable. The criteria should be stated at the beginning of the paper.

Author Response

Please see the attached document "Responses to Reviewers"

Reviewer 2 Report

It my be a good idea to add Heart Rate Variability to the use of Heart Rate (HR). The regulation of the heart rhythm, which has a very complex behaviour makes it reasonable to expect, that the regulation can only be described optimally by a method which has a diversity of different parameters describing partly different behaviours of the subsystems. Thus a method provides more prognostic information on patients after myocardial infarction. It has been shown that heart rate variation (HRV) provides more prognostic information on patients, for example after acute myocardial infarction than previously suggested risk markers

Author Response

(The authors gave the same response as above.)

Reviewer 3 Report

Please see attached PDF.

Author Response

(The authors gave the same response as above.)

Round 2

Reviewer 1 Report

In the latest version of the paper the authors considered most of my comments.

Reviewer 3 Report

None.